# Comparison of the Clinico-Microbiological Characteristics of Culture-Positive and Culture-Negative Septic Pulmonary Embolism: A 10-Year Retrospective Study

**DOI:** 10.3390/pathogens9120995

**Published:** 2020-11-27

**Authors:** Yoshito Nishimura, Hideharu Hagiya, Mikako Obika, Fumio Otsuka

**Affiliations:** Department of General Medicine, Okayama University Hospital, Okayama 7008558, Japan; hagiya@okayama-u.ac.jp (H.H.); obika-m@cc.okayama-u.ac.jp (M.O.); fumiotsu@md.okayama-u.ac.jp (F.O.)

**Keywords:** septic pulmonary embolism, blood culture, procalcitonin

## Abstract

Septic pulmonary embolism (SPE) is a rare yet serious infectious disorder with nonspecific clinical findings due to microorganism-containing emboli disseminating from extrapulmonary infectious foci. It is unknown whether a positive blood culture correlates with a worse clinical outcome. We compared the clinical and microbiologic characteristics of patients with SPE divided into the culture-positive group and the culture-negative one. This study was a retrospective observational study of the patients diagnosed with SPE and treated in an academic hospital from April 2010 to May 2020. We identified six culture-positive and four culture-negative patients with SPE during the study period. The culture-positive group had significantly longer periods of hospitalization (median: 75 days, range: 45–125 days) than the culture-negative group (median: 14.5 days, range: 3–43 days) (*p* < 0.05), as well as significantly elevated serum C-reactive protein and procalcitonin. Patients with culture-negative SPE more commonly had odontogenic infections as the primary infectious foci. Our study highlights the importance of giving extra attention to SPE patients who have a positive blood culture, as they may have worse clinical outcomes. Physicians need to collaborate with dentists when faced with patients with culture-negative SPE, since they may have primary odontogenic infections.

## 1. Introduction

Septic pulmonary embolism (SPE) is a rare systemic infectious disorder characterised by insidious onset and nonspecific clinical conditions, including fever, tachycardia, and chest pain [1,2,3,4,5]. In SPE, non-thrombotic microorganism-containing emboli disseminating from extrapulmonary primary infectious sources obstruct the small pulmonary vasculature, causing sepsis and multiple small lung abscesses [6]. The primary infectious foci include right-sided infective endocarditis [7,8], soft tissue infections, such as necrotising fasciitis [9,10], implantable device- and catheter-related infections [11,12,13], and odontogenic lesions [3,14]. Due to its nonspecific clinical presentations, diagnosis of SPE remains challenging for clinicians despite its high mortality [15].

While previous studies have accumulated data regarding risk factors for mortality and the need for critical care [15,16], detailed descriptions of patients with SPE with positive blood culture, which may be associated with a higher risk of mortality and severity [17], compared with the blood culture-negative patients have not been reported. Here, patients who had any positive culture results were defined as “culture-positive,” and those who did not have any positive culture throughout their hospital stays were described as “culture-negative”. This study aimed to clarify the detailed clinical and microbiological features of SPE in the light of blood culture positivity.

## 2. Results

### 2.1. Baseline Demographics

The baseline demographics and clinical characteristics of the patients are summarized in Table 1. Ten patients with SPE were identified from medical records during the study period. Six of these patients had a positive blood culture. The patients’ median age was 58.5 years (range: 31–79 years) and 70.5 years (range: 63–78 years) in the culture-positive and culture-negative groups, respectively. Three out of the ten patients had nosocomial diseases. Clinical symptoms on admission included fever (temperature higher than 38.0 °C), cough, shortness of breath, chest pain, hypotension (sBP less than 90 mm Hg), tachycardia (heart rate higher than 100/min), and tachypnea (respiratory rate higher than 22/min). The patients with positive blood cultures more commonly presented with fever, shortness of breath, hypotension, tachycardia, and tachypnea, although there were no statistically significant differences in the frequency between the two groups. Primary sources of SPE were identified as skin and soft tissue infections (two in the culture-positive group), left- and right-sided infective endocarditis (two and one in the culture-positive group, respectively), odontogenic infections (four in the culture-negative group), and central line-associated bloodstream infections (CLABSI) (one in the culture-positive group). The patients had various comorbidities, including diabetes and cardiovascular diseases, but there were no differences in the frequency of comorbidities between the groups. Regarding risk factors of infection, in the culture-positive group, one patient had a history of admission within six months of the diagnosis of SPE due to community-acquired pneumonia, one patient was considered immunocompromised due to the current use of corticosteroids and an anti-TNF-alpha inhibitor, and two patients had indwelling devices. No patient in the culture-negative group had such underlying risk factors.

### 2.2. Complications and Radiographic Findings

Chest CT revealed abnormalities, including bilateral infiltrates, nodular opacities, cavitation, and pleural effusion, in all the patients. Of note, all the patients had nodular opacities, while the culture-positive patients more frequently had pleural effusion. Regarding laboratory findings, serum C-reactive protein and procalcitonin levels were significantly higher in the culture-positive group than in the culture-negative group with a median of 18.1 mg/dL vs. 6.7 mg/dL and 1.67 ng/mL vs. 0.074 ng/mL, respectively (*p* < 0.05). In contrast, the serum sodium level was significantly lower in the culture-positive group (median: 133 mEq/L vs. 140 mEq/L, *p* < 0.05). Patients with SPE in the culture-positive group had higher white blood cell and D-dimer levels and lower potassium and chloride levels, although the differences were not significant. The radiographic findings and complications of the ten patients with SPE after admission are shown in Table 2. In-hospital complications included continuous bacteremia (three of the six culture-positive patients), drainage therapy (one in the culture-positive group), transfer to the intensive care unit (ICU) (four and one in the culture-positive and culture-negative groups, respectively). One patient in the culture-positive group died three days after admission (the patient was found to have cardiac arrest by a night shift nurse, and the attempt to resuscitate was unsuccessful). While the autopsy was deferred per the family’s request, the cause of death was considered SPE. The periods of hospitalization were significantly longer in the culture-positive group (median: 75 days, range: 45–125 days, excluding the one patient who died on hospital day 3) than in the culture-negative group (median: 14.5 days, range: 3–43 days) (*p* < 0.05).

### 2.3. Microbiologic Features and Treatments

Table 3 describes the detailed features of the patients with SPE, including microbiologic findings and antibiotic regimens. Blood cultures were obtained from every patient, and no culture-negative patients received antibiotics before collecting blood cultures. Isolated pathogens in the blood culture-positive group included methicillin-sensitive *Staphylococcus aureus* (MSSA), *Staphylococcus capitis*, and *Staphylococcus lugdunensis* in the patient with infective endocarditis, *Fusobacterium nucleatum* and *Campylobacter rectus* in the patient with clival osteomyelitis, and *Candida albicans* and *Candida glabrata* in the patient with a CLABSI. Except for one patient who refused intravenous antibiotics against medical advice (Case No. 7), all the patients received parenteral antimicrobial therapy. According to the antimicrobial sensitivity results, three patients underwent antimicrobial deescalation. Due to the disease severity, five of them required ICU transfer during the admission because of hypotension requiring vasopressors. Although all the patients transferred to the ICU had prolonged hospital stays (median: 80.5 days), they were discharged without subsequent complications.

## 3. Discussion

This study retrospectively examined the baseline and clinical characteristics of patients with SPE, focusing on the patients with positive blood cultures. SPE is a severe yet rare infectious disorder that is hard to recognize. It is partly because of the non-specificity of the associated symptoms such as fever, cough, and shortness of breath, as shown in the current study and previous studies [1,5,16].

Negative blood cultures have been reported in 30–50% of patients with sepsis [18]. Counterintuitively, whether patients with blood culture-positive sepsis have poorer clinical outcomes than their blood culture-negative counterparts has been controversial. In 2019, Sigakis and colleagues reported that culture-negative sepsis patients had similar clinical characteristics as well as similar rates of mortality to culture-positive cases in a retrospective case–control study [19]. In the meantime, two other studies reported that culture-positive sepsis patients had significantly higher mortality and worse severity of illness [18,20]. Likewise, it has been undetermined if patients with culture-positive SPE may have poorer clinical outcomes, as well as increased mortality. A Korean retrospective study suggested that tachypnea and segmental or lobar consolidation on the chest CT scan, not a positive blood culture, were independent risk factors for in-hospital death in patients with SPE [21]. A Chinese retrospective analysis implied that hypotension and ineffective or delayed antimicrobial therapy were independent risk factors for mortality [16]. Although the two previous studies did not suggest a positive blood culture as a risk factor for in-hospital death or clinical exacerbation, the studies did not perform a statistical analysis to look for relevance between blood culture positivity and worse clinical outcomes. In our study, culture-positive patients were more commonly transferred to the ICU and had significantly longer periods of hospitalisation, as well as elevated serum inflammatory markers. It could be partly due to a higher bacterial burden, as suggested previously in the studies of culture-positive and -negative sepsis [20,22]. A future study combining genomic bacterial load measurement would give us further insights into the difference in the severity of culture-positive and -negative SPE. It is also worth noting that patients in the culture-positive group had significantly lower serum sodium levels than in the culture-negative group, as even mild hyponatremia is associated with increased mortality [23]. Due to the small number of eligible cases, the prognosis of the patients could not be compared appropriately in terms of blood culture positivity. Based on the previous findings related to positive blood culture and our results, clinicians need to vigorously perform blood culture sampling in parallel with searching for the primary sources of infection and giving empiric antibiotic treatments for patients suspected of having SPE.

In our study, all the patients with culture-negative SPE had odontogenic infections, including periodontitis, dental caries, or odontogenic maxillary sinusitis as the primary infectious foci. A previous study suggested that SPE due to a periodontal disease might have a milder disease course than in patients with SPE with non-odontogenic sources of infections [3]. Combined with the implications of a positive blood culture and disease severity, our findings imply that patients with SPE with odontogenic infections may have less a severe disease and a negative blood culture. In the meantime, physicians need to pay attention to odontogenic infections, liaising closely with dentists when faced with patients with culture-negative SPE.

Our study has a few limitations that should be considered upon reviewing the results. First, we performed the study at a single Japanese university hospital, which reduces the generalizability of the results to SPE patients from other countries and ethnicities. Secondly, due to the rarity of SPE, we could include only ten patients during the 10-year study period. Because of the small number of participants, this study may have a decreased power to detect significant associations between variables. A future multicenter prospective cohort study is warranted to overcome these limitations. The possibility of underreporting of the cases in the electronic health records should also be noted as a factor for the small number of patients.

Despite the limitations, our retrospective study provides valuable data to elaborate upon the clinical and microbiologic characteristics of patients with SPE. Due to the severity of the disease and the lack of characteristic symptoms, SPE should be suspected in patients with various infections who have multiple nodular lesions on the chest CT scan. Physicians should give extra attention to SPE patients with a positive blood culture, as they may have worse clinical outcomes requiring prolonged periods of hospitalisation.

## 4. Materials and Methods

### 4.1. Design and Study Population

In this retrospective observational study, we reviewed patients’ electronic health records at the Okayama University Hospital, an 865-bedded tertiary care national university hospital in Japan, from 1 April 2010, to 31 May 2020 (10 years). To identify cases of SPE, we used a computer-assisted search, with the following data collected: age, sex, clinical symptoms on admission, classification of onset (community or nosocomial onset), clinical courses, primary infection foci, comorbidities, radiological findings, laboratory results, and microbiological findings. The classification of onset was defined as nosocomial if the patient had signs of infection more than 48 h after their admission. The diagnostic criteria of SPE suggested by Cook and colleagues [1] was used to identify the cases. In brief, the included patients met the following four criteria: (1) a chest computed tomography (CT) scan showing focal or multifocal lung infiltrates compatible with SPE; (2) presence of an active extrapulmonary infection as a pulmonary embolic source; (3) exclusion of other causes of lung infiltrates; and (4) resolution of lung lesions with antibiotic therapy or death. All the patients had equal or more than two systemic inflammatory response syndrome (SIRS) criteria with a suspected source of infection. All patients less than 18 years old were excluded.

### 4.2. Definitions

We defined vital sign abnormalities as follows: hypotension, systolic blood pressure (sBP) below 90 mm Hg; tachycardia, heart rate higher than 100/min; and tachypnea, the respiratory rate higher than 22 breaths/min. Immunocompromised conditions were defined as current corticosteroid and/or biologic use. Patients who had any positive culture results were defined as “culture-positive,” and those who did not have any positive culture throughout their hospital stays were defined as “culture-negative”. Continuous bacteriemia was defined as a positive blood culture for more than three days after the initial appropriate antibiotic administration.

### 4.3. Statistical Analysis

We used the Wilcoxon and Kruskal–Wallis tests to compare the continuous variables between the blood culture-positive and blood culture-negative groups. The Fisher’s exact test was used to compare the categorical data between the two groups. The threshold for significance was defined as the *p*-value < 0.05. All the statistical analyses were conducted with JMP Version 15.1 (SAS Institute, Cary, NC, USA).

### 4.4. Ethics Approval

The study protocol was approved by the institutional review board of the Okayama University Hospital (reference number 2008-005, approved on 28 August 2020).

## Figures and Tables

**Table 1 pathogens-09-00995-t001:** Baseline demographics and clinical characteristics of the SPE patients.

	n (%)		*p*-Value
	Culture-Positive (n = 6)	Culture-Negative (n = 4)	
Age (years), median (range)	58.5 (31–79)	70.5 (63–78)	0.456
Male	4 (66.7)	3 (75.0)	1.000
Clinical symptoms on admission			
Fever (BT > 38.0 ℃)	4 (66.7)	1 (25.0)	0.262
Cough	1 (16.7)	1 (25.0)	0.867
Shortness of breath	3 (50.0)	1 (25.0)	0.452
Chest pain	1 (16.7)	2 (50.0)	0.967
Weight loss	0	1 (25.0)	1.000
Hypotension (sBP < 90 mm Hg)	2 (33.3)	0	0.333
Tachycardia (HR > 100/min)	4 (66.7)	1 (25.0)	0.262
Tachypnea (RR > 22/min)	5 (83.3)	1 (25.0)	0.120
Primary source of SPE			
Skin and soft tissue infections	2 (33.3)	0	0.333
Left-sided IE	2 (33.3)	0	0.333
Right-sided IE	1 (16.7)	0	0.600
Odontogenic infections	0	4 (100)	0.048 *
CLABSI	1 (16.7)	0	0.600
Comorbidity			
Diabetes	0	1	1.000
Cardiovascular diseases	3 (50.0)	0	0.167
Venous thromboembolism	1 (16.7)	1 (25.0)	0.867
Rheumatologic diseases	1 (16.7)	0	0.600
Cerebrovascular diseases	1 (16.7)	1 (25.0)	0.867
Dental caries	1 (16.7)	2 (50.0)	0.967
Risk factors			
Admission within 6 months	1 (16.7)	0	0.600
Immunocompromised	1 (16.7)	0	0.600
Indwelling devices	2 (33.3)	0	0.333

* Statistically significant difference between the two groups (*p* < 0.05). Abbreviations: BT, body temperature; BP, blood pressure; CLABSI, central line-associated bloodstream infection; HR, heart rate; IE, infective endocarditis; RR, respiratory rate; SPE, septic pulmonary embolism.

**Table 2 pathogens-09-00995-t002:** In-hospital complications and radiographic characteristics of the SPE patients.

	Median (Range)		*p*-Value
	Culture-Positive (n = 6)	Culture-Negative (n = 4)	
In-hospital complications			
Continuous bacteremia, n (%)	3 (50.0)	N/A	
Drainage, n (%)	1 (16.7)	0	0.600
ICU transfer, n (%)	4 (66.7)	1 (25.0)	0.262
In-hospital 30-Day mortality, n (%)	1 (16.7)	0	0.867
Lengths of hospital stay ^1^, days	75 (45–125)	14.5 (3–43)	0.012 *
CT findings			
Bilateral infiltrates, n (%)	4 (66.7)	3 (75.0)	0.833
Nodular opacities, n (%)	6 (100)	4 (100)	-
Cavitation, n (%)	2 (33.3)	1 (25.0)	0.667
Pleural effusion, n (%)	4 (66.7)	1 (25.0)	0.262
Laboratory findings			
WBC (10^3^/µL)	10.21 (0.61–13.4)	8.31 (5.84–22.8)	0.643
Serum CRP (mg/dL)	18.1 (8.9–34.6)	6.7 (1.2–7.6)	0.020 *
Serum PCT (ng/mL)	1.67 (0.61–93)	0.074 (0.021–0.38)	0.020 *
Serum sodium (mEq/L)	133 (132–136)	140 (137–146)	0.013 *
D-dimer (µg/mL)	2.9 (0.50–6.0)	1.3 (0.60–3.2)	0.366

* Statistically significant difference between the two groups (*p* < 0.05). Abbreviations: CT, computed tomography; CRP, C-reactive protein; ICU, intensive care unit; PCT, procalcitonin; WBC, white blood cells. ^1^ Because one patient in the culture-positive group died on hospital day 3, the length of hospital stay in the culture-positive group is the median (range) of the other five patients.

**Table 3 pathogens-09-00995-t003:** Detailed demographics, clinical characteristics, and outcomes of the SPE patients.

Case No.	Age/Sex	Onset	Chief Complaint	Underlying Conditions	Primary Infection	Culture Sources	Pathogens	Antibiotics	Treatment Length (Day)	ICU Transfer	LOS (Days)
1	75/M	Community	Suspected lung cancer	AS S/P mechanical valve placement	Left-sided IE Odontogenic	Blood	*S. capitis*	CTRX	3	-	3 (death)
2	34/M	Community	Fever, headache	TOF	Right-sided IE	Blood	MSSA	LVFX + DAP CEZ + GM	125	Yes	125
3	31/M	Nosocomial	Productive cough	TA/PA/MAPCAs	Left-sided IE	Blood	*S. lugdunensis*	MEPM + VCM CEZ	36	Yes	45
4	70/F	Nosocomial	Fever, headache		Clival osteomyelitis	Blood	*F. nucleatum*, *C. rectus* (blood)	MEPM + MNZ CEZ + MNZ	50	-	57
5	79/F	Community	Fever		Vertebral osteomyelitis	Blood	MSSA (blood)	CEZ	48	Yes	86
6	47/M	Nosocomial	Fever, chest pain		CRBSI	Blood, catheter tip	*C. albicans*, *C. glabrata* (blood, catheter tip)	MCFG + FLCZ	50	Yes	75
7	76/M	Community	Fever, cough		Odontogenic	Blood, BALF	None (blood) *S.intermedius*, *Fusobacterium spp.* (BALF)	AMPC/CVA, oral	14	-	3
8	63/F	Community	Fever, back pain		Odontogenic sinusitis	Blood	None	ABPC/SBT	14	-	14
9	65/M	Community	Fever, AMS		Odontogenic	Blood	None	MEPM + TEIC	29	Yes	43
10	78/M	Community	Fever		Odontogenic	Blood	None	ABPC/SBT	14	-	15

Abbreviations: ABPC/SBT, ampicillin/sulbactam; AMPC/CVA, amoxicillin/clavulanate; AMS, altered mental status; AS, aortic stenosis; BALF, bronchoalveolar lavage fluid; CEZ, cefazoline; CTRX, ceftriaxone; CRBSI, catheter-related bloodstream infection; DAP, daptomycin; FLCZ, fluconazole; GM, gentamycin; IE, infective endocarditis; LOS, lengths of stay; LVFX, levofloxacin; MAPCAs, major aortopulmonary collaterals; MCFG, micafungin; MEPM, meropenem; MNZ, metronidazole; MRSA, methicillin-resistant *Staphylococcus aureus*; MSSA, methicillin-sensitive *Staphylococcus aureus*; PA, pulmonary atresia; S/P, status post; TA, tricuspid atresia; TEIC, teicoplanin; TOF, tetralogy of Fallot; VCM, vancomycin.

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
