# Peer review of "Comparison of the Clinico-Microbiological Characteristics of Culture-Positive and Culture-Negative Septic Pulmonary Embolism: A 10-Year Retrospective Study"

_pathogens, 2020, doi:10.3390/pathogens9120995_

Round 1
Reviewer 1 Report
The manuscript by Nishmura et al reports the major clinical differences of culture-positive and culture-negative SPE in lengths of hospitalization, elevated serum inflammatory markers as well as odontogenic infections by a 10-year retrospective study of 10 SPE patients. This is a very interesting funding showing the correlation of blood culture and clinical outcome, which rises our attention on the significance of blood culture for SPE patients. Importantly, the study firstly reveals that culture-positive patients have higher serum C-reactive protein and procalcitonin and even longer hospitalization and treatment. Besides, the report also demonstrates more common odontogenic infection exclusively in culture-negative patients, indicating the necessity of dentist involvement on corresponding SPE patients. Overall, the study is well-organized in a logical manner, and the report has some significances for the clinical prognosis and treatment.
Major points:
- It is still unclear to know why culture-positive patients are more prone to have higher serum inflammatory markers and even longer hospitalization and treatment, though the authors have made connections between blood culture and its clinical characteristics. An explanation on its correlation will be needed for people to have scientific understanding on its causality.
- I would still say that the number of SEP patients is too limited which weaken the reliable statistical analysis in this study even though the authors mentioned in the Discussion part.
Minor points:
The authors should explain what is blood culture for SEP patients, and what kind of results represent “Positive” and “Negative” in Introduction part. It is a very important term in this manuscript and the introduction will make it easier to understand for readers without related background.
Author Response
Comment from Reviewer 1
- Comment 1: [It is still unclear to know why culture-positive patients are more prone to have higher serum inflammatory markers and even longer hospitalization and treatment, though the authors have made connections between blood culture and its clinical characteristics. An explanation on its correlation will be needed for people to have scientific understanding on its causality.]
Response: Thank you for reviewing our manuscript. We speculate that the reason why culture-positive patients were more likely to have higher levels of inflammatory markers and longer hospitalization could be due to higher bacterial burden as noted previously (Lisboa T, et al. Crit Care Med. 2010 Oct;38(10 Suppl:S656-62. doi: 10.1097/CCM.0b013e3181f2453a). In the future, as noted in the article above, it would give us further insights into pathophysiology of SPE if we could measure genomic bacterial load. We have added the arguments in the Discussion section.
- Comment 2: [I would still say that the number of SEP patients is too limited which weaken the reliable statistical analysis in this study even though the authors mentioned in the Discussion part.]
Response: Thank you for your comment. We duly understand the limitation as you could mention. We believe further analysis with multicenter prospective cohort study which we are currently planning, would be warranted to overcome the limitation. We specifically noted the need for a future multicenter prospective cohort study in the manuscript.
- Comment 3: [The authors should explain what is blood culture for SEP patients, and what kind of results represent “Positive” and “Negative” in Introduction part. It is a very important term in this manuscript and the introduction will make it easier to understand for readers without related background.]
Response: We have clarified the definition of “culture-positive” and “culture-negative” in Introduction as well as Materials and Methods parts. Also, we noted that all patients in the study had ≥ 2 SIRS criteria with suspected sources of infection in Material and Methods part.

Reviewer 2 Report
I am glad having the opportunity of reviewing the Manuscript 1007733 entitled ‘A Comparison of the Clinical Characteristics of Culture-Positive and Culture-Negative Septic Pulmonary Embolism: A 10-Year Retrospective Study’ for ‘Pathogens.
I believe some alterations and improvements are needed.
My comments:
- Title: ‘A Comparison of the Clinical Characteristics of Culture-Positive and Culture-Negative Septic Pulmonary Embolism’: please rephrase as your manuscript does not refer only to clinical characteristics.
- Abstract: ‘A single paragraph of about 200 words maximum’. Please delete comment.
- Abstract: ‘a rare yet serious infectious disease’, please instead of disease use ‘disorder’ or ‘complication’
- Abstract: ‘…and microorganism-containing emboli’: please instead of ‘and’ use ‘due to …’
- Introduction:
- ‘While previous studies have accumulated data regarding risk factors for mortality and the need for critical care, detailed descriptions of patients with SPE with positive blood culture, which may be associated with a higher risk of mortality and severity, compared with the blood culture-negative patients have not been reported’: So according to your statement, the reader waits to read something about mortality as a clinical outcome.
- Results - Baseline Demographics
- ‘Regarding risk factors of infection, in the culture-positive group, one patient had a history of admission within six months of the diagnosis of SPE’: please state the reason of hospitalization of this patient, for example due to infection of
- ‘due to the use of corticosteroids and biologics’: please explain what biologics refers to.
- Table 1 contains both Baseline Demographic and clinical Characteristics. Please rephrase or split Table 1 to two Tables.
- ‘The lengths of hospitalisation were significantly longer’: correct to ‘The length of hospitalisation was significantly longer..
- Please describe CT findings prior to clinical outcome
- How do you explain that serum sodium level was significantly lower in the culture-positive group? Why do you believe this is important and you should refer to in your paper? Please explain for potassium and chloride also.
- ‘were more prone to higher white… ‘: please use instead ‘had higher’
- Table 2. ‘Radiographic Characteristics of the SPE Patients’: please rephrase the legend as Table 2 includes not only radiographic Characteristics.
- Table 3. ‘Detailed Features of the SPE Patients’. Please rephrase the legend as Table 3 includes also demographics, Clinical Characteristics, previous medical history, outcome;
- Clinical presentations of SPE range from insidious illness with mild respiratory symptoms to respiratory failure and septic shock. What was the main reason for admission of your patients in the ICU?
- Why the patient who eventually died have not been transferred to ICU? What was the cause of death?
- How is possible that the patient No 6 being in a severe medical situation justifying ICU admission has not received parenteral antibiotics?
- Please make a statement about mortality as a clinical outcome in 'Results'
Discussion
- Immunocompromised conditions were defined as prior or current corticosteroid and/or biologic use?
- ‘This study retrospectively examined the baseline and clinical characteristics of patients with SPE in Japan’: This is a single-center study, not a national one. Please omit in Japan.
- Minor English language polishing is needed throughout the text.
Author Response
Comment from Reviewer 2
- Comment 1: [Title: ‘A Comparison of the Clinical Characteristics of Culture-Positive and Culture-Negative Septic Pulmonary Embolism’: please rephrase as your manuscript does not refer only to clinical characteristics.]
Response: Thank you for your precious time to review our manuscript. We have changed the title to “A Comparison of the Clinico-microbiological Characteristics…” according to your suggestion.
- Comment 2: [Abstract: ‘A single paragraph of about 200 words maximum’. Please delete comment. / Abstract: ‘a rare yet serious infectious disease’, please instead of disease use ‘disorder’ or ‘complication’ / Abstract: ‘…and microorganism-containing emboli’: please instead of ‘and’ use ‘due to …’]
Response: Thank you for the suggestion. We have revised the manuscript accordingly.
- Comment 3: [Introduction:‘While previous studies have accumulated data regarding risk factors for mortality and the need for critical care, detailed descriptions of patients with SPE with positive blood culture, which may be associated with a higher risk of mortality and severity, compared with the blood culture-negative patients have not been reported’: So according to your statement, the reader waits to read something about mortality as a clinical outcome.]
Response: We included “In-hospital 30-Day Mortality” in the manuscript which was noted in Table 2. The mortality was 16.7% in the Culture-Positive group and no one passed in the Culture-Negative group.
- Comment 4: [Results - Baseline Demographics‘Regarding risk factors of infection, in the culture-positive group, one patient had a history of admission within six months of the diagnosis of SPE’: please state the reason of hospitalization of this patient, for example due to infection of]
Response: As the patient was admitted due to community-acquired pneumonia, we have added the information in the text.
- Comment 5: [‘due to the use of corticosteroids and biologics’: please explain what biologics refers to]
Response: It refers to an anti-TNF-alpha inhibitor, and we have added the information in the text as well.
- Comment 6: [Table 1 contains both Baseline Demographic and clinical Characteristics. Please rephrase or split Table 1 to two Tables.]
Response: We have rephrased the title of Table 1 as “Baseline Demographics and Clinical Characteristics of the SPE Patients”. Also, Table 2 was rephrased as “In-hospital Complications and Radiographic Characteristics of the SPE Patients”.
- Comment 7: [‘The lengths of hospitalisation were significantly longer’: correct to ‘The length of hospitalisation was significantly longer..]
Response: We have revised the manuscript according to your suggestion.
- Comment 8: [Please describe CT findings prior to clinical outcome]
Response: We have reversed the order of CT findings and complications in the main text accordingly.
- Comment 9: [How do you explain that serum sodium level was significantly lower in the culture-positive group? Why do you believe this is important and you should refer to in your paper? Please explain for potassium and chloride also.]
Response: We assume that hyponatremia in the culture-positive group could be multifactorial including due to SIADH with severe sepsis, relative hypocortisonism associated with severe infection, and poor oral intake. As hyponatremia is known to be associated with increased mortality (Waikar SS et al. Am J Med. 2009 Sep; 122(9): 857–865.), we believe it is worth referring to the serum sodium level in the text and we have added the argument in the Discussion section. As we have not mentioned the difference in potassium and chloride in the text, and we do not think there are clinically meaningful pertinent findings, we have omitted the potassium/chloride levels from the Table 2.
- Comment 10: [‘were more prone to higher white… ‘: please use instead ‘had higher’ / Table 2. ‘Radiographic Characteristics of the SPE Patients’: please rephrase the legend as Table 2 includes not only radiographic Characteristics. / Table 3. ‘Detailed Features of the SPE Patients’. Please rephrase the legend as Table 3 includes also demographics, Clinical Characteristics, previous medical history, outcome;]
Response: We have revised the manuscript accordingly. The legend of Table 2 is rephrased as“In-hospital Complications and Radiographic Characteristics of the SPE Patients”. The legend of Table 3 is now “Detailed Demographics, Clinical Characteristics and Outcome Features of the SPE Patients”.
- Comment 11: [Clinical presentations of SPE range from insidious illness with mild respiratory symptoms to respiratory failure and septic shock. What was the main reason for admission of your patients in the ICU?]
Response: All the patients who were upgraded to the ICU required vasopressors due to persistent hypotension. We have added the reason for the ICU transfer in the Results section.
- Comment 12: [Why the patient who eventually died have not been transferred to ICU? What was the cause of death?]
Response: The patient was found dead by a night-shift nurse, and the attempt to resuscitate was unsuccessful. We have added the information in the text. While autopsy was deferred per the family’s request, the cause of death was considered to be due to SPE at the time.
- Comment 13: [How is possible that the patient No 6 being in a severe medical situation justifying ICU admission has not received parenteral antibiotics?]
Response: The patient who did not receive IV antibiotics was the patient No. 7 (It was noted as No. 6 due to typo in the text). We apologize for the confusion. Patient No. 7 did not receive IV antibiotics due to the patient’s refusal against medical advice. We have added the information in the text.
- Comment 14: [Please make a statement about mortality as a clinical outcome in 'Results']
Response: Thank you for your suggestion. We already have “In-hospital 30-Day Mortality” in Table 2. Also, no other patients have passed after discharge.
- Comment 15: [Discussion: Immunocompromised conditions were defined as prior or current corticosteroid and/or biologic use?]
Response: Thank you for your comment. We defined immunocompromised conditions as “current corticosteroid and/or biologic use.” We have added the word “current” in the section 4.2 Definitions, in the Materials and Methods.
- Comment 16: [‘This study retrospectively examined the baseline and clinical characteristics of patients with SPE in Japan’: This is a single-center study, not a national one. Please omit in Japan.]
Response: We have deleted “in Japan” from the sentence.
- Comment 17: [Minor English language polishing is needed throughout the text.]
Response: We used Editage professional proofreading service before submission. To further improve the quality of English, we asked them to proofread the manuscript again before submitting revision.

Round 2
Reviewer 2 Report
Dear Authors,
I am glad having the opportunity of reviewing the Manuscript 1007733 entitled ‘A Comparison of the Clinical Characteristics of Culture-Positive and Culture-Negative Septic Pulmonary Embolism: A 10-Year Retrospective Study’ for ‘Pathogens.
I am glad to inform you that after revision the Manuscript is greatly improved.